# Study protocol for the BUSCopan in LABor (BUSCLAB) study: A randomized placebo-controlled trial investigating the effect of butylscopolamine bromide to prevent prolonged labor

Ingvil Krarup Sørbye[1]*, Lise Christine Gaudernack[1,2], Angeline Einarsen[1], Leiv Arne Rosseland[3,4], Mirjam Lukasse[2,5], Nina Gunnes[6], Trond Melbye Michelsen[1,4]

1 Division of Obstetrics and Gynecology, Department of Obstetrics, Oslo University Hospital, Oslo, Norway, 2 Faculty of Health Sciences, Department of Nursing and Health Promotion, Oslo Metropolitan University, Oslo, Norway, 3 Division of Emergencies and Critical Care, Department of Research and Development, Oslo University Hospital, Oslo, Norway, 4 Faculty of Medicine, Institute of Clinical Medicine, University of Oslo, Oslo, Norway, 5 Department of Nursing and Social Sciences, University of South-Eastern Norway, Notodden, Norway, 6 Norwegian Research Centre for Women's Health, Oslo University Hospital, Oslo, Norway

* ingvil.sorbye@ous-hf.no

**Data Availability Statement:** No datasets were generated of analysed during the current study. All

## Abstract

### Background

First-time mothers are prone to prolonged labor, defined as the crossing of partograph alert or action lines. Prolonged labor may occur among as many as one out of five women, and is associated with a range of adverse birth outcomes. Oxytocin is the standard treatment for prolonged labor, but has a narrow therapeutic window, several adverse effects and limited efficacy. Despite poor evidence, labor wards often use antispasmodic agents to treat prolonged labor. The antispasmodic drug butylscopolamine bromide (Buscopan®) may shorten duration of labor, but studies on prevention of prolonged labor are lacking. In this randomized double-blind placebo-controlled clinical trial, we aim to evaluate the effect of butylscopolamine bromide on duration of labor in first-time mothers showing first signs of slow labor progress by crossing the World Health Organization partograph alert line.

### Methods and analysis

The study is a single center study at Oslo University Hospital, Oslo, Norway. We will recruit 250 primiparous women with spontaneous labor start at term. Women are included in the first stage of labor if they show signs of slow labor progress, defined as the crossing of the partograph alert line with a cervical dilation between 3–9 cm. Participants are randomized 1:1 to either 20 mg intravenous butylscopolamine bromide or intravenous placebo (1 mL sodium chlorine 9 mg/mL). We considered a mean difference of 60 minutes in labor duration clinically relevant. The primary outcome is duration of labor from the provision of the investigational medicinal product to vaginal delivery. The secondary outcomes include change in

relevant data from this study will be made available upon study completion.

**Funding:** The BUSCLAB study is funded by grants from Oslo University Hospital (https://oslo-universitetssykehus.no) (PhD grant LG), Stiftelsen Dam (https://dam.no)/the Norwegian Women's Public Health Association (https://sanitetskvinnene.no) (PhD grant number 2020/FO283405) to AE), the Norwegian Midwifery Association (https://jordmorforeningen.no) (TM), and the Royal Norwegian Society of Sciences and Letters (https://dknvs.no) (TM). The funders had and will not have a role in study design, data collection and analysis, decision to publish, or preparation of the manuscript.

**Competing interests:** The authors have declared that no competing interests exists.

labor pain, use of oxytocin augmentation, delivery mode, and maternal birth experience. The primary data for the statistical analysis will be the full analysis set and will occur on completion of the study as per the prespecified statistical analysis plan. The primary outcome will be analyzed using Weibull regression, and we will treat cesarean delivery as a censoring event.

## Introduction

Prolonged labor is a common condition, and occurs among one out of five women [1]. Longer labor in modern obstetric cohorts is associated with primiparity, high maternal age and high maternal BMI [2]. Prolonged labor, often defined as the crossing of partograph action lines, is associated with several undesired outcomes, such as a negative birth experience, operative delivery, chorioamnionitis, and infant admission to intensive care [3–6]. Two large studies on healthy first-time mothers' labor duration have found a mean dilation rate in the active phase of labor of 1.2 cm and 1.4 cm per hour, respectively [7,8]. When progress in the active phase of labor (regular contractions and cervical dilation of 3 cm or more) is too slow, augmentation with oxytocin is indicated. However, use of oxytocin is not without risk of adverse effects. Oxytocin has been described as the drug most commonly related to preventable adverse events during labor and has a very unpredictable therapeutic index [9]. Use of oxytocin to stimulate contractions is associated with an increased risk of uterine hyperstimulation, which may lead to compromised oxygen supply to the fetus, fetal asphyxia [10–12], and the need for immediate operative delivery [12,13]. Augmentation with oxytocin in first labor has also shown associations with anal sphincter injuries [14], postpartum hemorrhage and urinary retention [15,16], a negative birth experience [17] and possibly delayed initiation of breastfeeding [18].

Studies on the prevalence of augmentation with oxytocin report a high and increasing use, especially among first-time mothers [13,19,20]. A nationwide study in Norway during the period 2000–2011 found that 44%–48% of primiparous women in spontaneous labor at term received oxytocin augmentation [21]. Given that there is uncertainty regarding the efficacy of oxytocin and that the use of oxytocin has considerable adverse effects, there is a need to evaluate alternative or adjuvant treatments for slow progress in the first stage of spontaneous labor.

Another class of drugs, antispasmodics, are used to prevent poor progress of labor in both high-income and low- and middle-income countries [22]. Antispasmodics are drugs that relieve spasms of smooth muscle tissue by either a musculotropic or neurotropic effect. Musculotropic antispasmodics act directly to relax and inhibit spasm of smooth muscle whilst having a mild calcium channel blocking effect, but no anticholinergic effects [23]. Neurotropic antispasmodics, on the other hand, break the connection between the parasympathetic nerve and the smooth muscle, acting as antagonists of acetylcholine at muscarinic receptors, thus also inhibiting muscle spasm. Both classes of drugs can in theory affect the uterine cervical dilation in labor as the cervix is partly composed of smooth muscle tissue and is innervated by parasympathetic nerve fibers [23,24].

Butylscopolamine bromide (Buscopan®) is an antispasmodic and anticholinergic drug, which binds to and blocks muscarinic receptors located on parasympathetic nerve endings and on smooth muscle cells. Despite a lack of robust evidence, the use of butylscopolamine bromide is quite common in labor wards to treat poor progress in labor. Some side effects have been reported for butylscopolamine bromide; patients sometimes experience tachycardia and dryness of the mouth. Information provided by the pharmaceutical industry mentions

that if given near delivery, the fetal heart rate might be influenced [25]. A possible influence on the fetal heart rate is likely to be caused by the temporary tachycardia experienced by the mother. Unlike scopolamine, butylscopolamine bromide does not cross the blood-brain barrier [26]. In line with this, although in vitro studies have demonstrated an interaction between butylscopolamine and the placental choline transport system, butylscopolamine bromide is not thought to cross the placenta [27]. The drug has been used in previous randomized controlled trials (RCTs) in order to facilitate cervix dilation and thus shorten duration of labor. A Cochrane review from 2013 including 21 RCTs on the use of antispasmodics found a reduction in the duration of first stage of labor and an increase in the cervical dilation rate in women receiving antispasmodics compared to placebo; however, evidence was of low quality [22]. No effect was found on the duration of the second and third stages of birth. None of the 17 studies included in the Cochrane review reported an increase in birth asphyxia nor an increase in required resuscitation or admission to the neonatal care unit. Following this Cochrane report, another study found a reduction in duration of active labor for first-time mothers of 57 minutes when treated with butylscopolamine bromide as soon as the active stage of labor started [28]. This study included 382 women randomized to 1 mL (20 mg) butylscopolamine bromide or 1 mL saline in active labor. A recent systematic review by Riemma et al. included eight randomized controlled trials in primiparous women with spontaneous start of labor, and found that butylscopolamine bromide shortened the average duration of the active phase of labor with 55.09 minutes (95% CI -68.83, -41.35) [29]. However, none of the included studies used antispasmodics to treat slow progress of labor and none of them were performed in high-income countries. The reviews have concluded that larger, rigorous RCTs are needed to evaluate the effect of antispasmodics on labor [22,29]. Our BUSCopan in Labor (BUSCLAB) study responds to this need. The aim here is to describe the design of the BUSCLAB trial and provide a rationale for the design elements incorporated into the study.

## Hypothesis and objective

The BUSCLAB study aims to assess the effect of butylscopolamine bromide on duration of labor from administration of the investigational medicinal product (IMP) to vaginal delivery in first-time mothers, who show first signs of slow labor progress. The null hypothesis is that there is no difference in the duration of labor from IMP administration to delivery between the two treatment groups.

## BUSCLAB trial design

The study is an interventional phase 3 double-blind two-arm randomized placebo-controlled trial to investigate the effect of intravenously administered butylscopolamine bromide compared with placebo on the duration of labor in first-time mothers who cross the alert line for labor dystocia, according to the World Health Organization (WHO) partograph.

## Methods: Participants, interventions, and outcomes

### Study setting

The study is conducted at a single tertiary birth center at Oslo University Hospital, Oslo, Norway. Primiparous women booked at Oslo University Hospital Rikshospitalet for a planned vaginal delivery receive an invitation to participate in the study six weeks before their due date. Women who have received and understood the information about the study and are willing to participate may be included. They sign the informed consent form when they arrive at the hospital to give birth.

## Eligibility criteria

Primiparous women aged 18 years or older with spontaneous start of labor and with a fetus in the vertex presentation at term are eligible for the study if they show first signs of slow labor progress. This is defined as crossing the alert line of the WHO partograph [30]. The WHO partograph has two diagonal lines: an alert line and an action line. The alert line goes from 3 to 10 cm and corresponds to an average dilation rate of 1 cm per hour. If the labor curve crosses to the right of this alert line, the dilation is less than 1 cm per hour in the active phase of first stage of labor. In the present study, we will include women at risk of having a prolonged labor defined as those with a dilation rate of less than 1 cm per hour (i.e., crossing the WHO partograph alert line but not the action line) in the active phase of labor with a cervix dilation $\geq 3-< 10$ cm). The schedule of enrollment, intervention and assessment is displayed in Fig 1.

The BUSCLAB trial design with inclusion/exclusion criteria is shown in Fig 2.

| TIME | Screening Period | STUDY PERIOD | | | | | |
|---|---|---|---|---|---|---|---|
| | | Delivery | | | End of postnatal stay | | Follow-up evaluation |
| Timepoint | | Baseline (≥3cm and < 10 cm dilation) Crossing the alert line | IMP is given | 30 min after IMP is given | Day of delivery | Day of discharge | 1 month postpartum |
| **ENROLMENT:** | | | | | | | |
| Informed consent | X | | | | | | |
| Inclusion/exclusion Evaluation | X | | | | | | |
| Medical History | X | | | | | | |
| Concomitant Medication | X | X | | | | | |
| [a]Physical Examination fetus (CTG) | | X | X | X | | | |
| [b]Vital signs mother | | X | X | | X | X | |
| Cervical dilation | | X | | | | | |
| Pain measurement (VAS) | | | X | X | | | |
| **INTERVENTIONS:** | | | | | | | |
| Treatment administration | | | X | | | | |
| **ASSESSMENTS:** | | | | | | | |
| Adverse event, mother | | | X | X | X | X | |
| Adverse event, newborn/fetus | | | X | X | X | X | |
| [c]Physical Examination newborn (pediatrician) | | | | | | 1st day post partum | |
| Childbirth Experience Questionnaire | | | | | | | X |

**Fig 1. Schedule of enrollment, intervention and assessments.** IMP = Investigational Medicinal Product. CTG = Cardiotocography—continuous fetal heart tracing. VAS = Visual Analogue Scale. [a]Physical Examination fetus includes fetal position and CTG—continuous fetal heart tracing [b]Vital signs include maternal blood pressure and heart rate. Height and body weight are obtained from pregnancy charts [c]Physical Examination newborn includes an examination of general appearance.

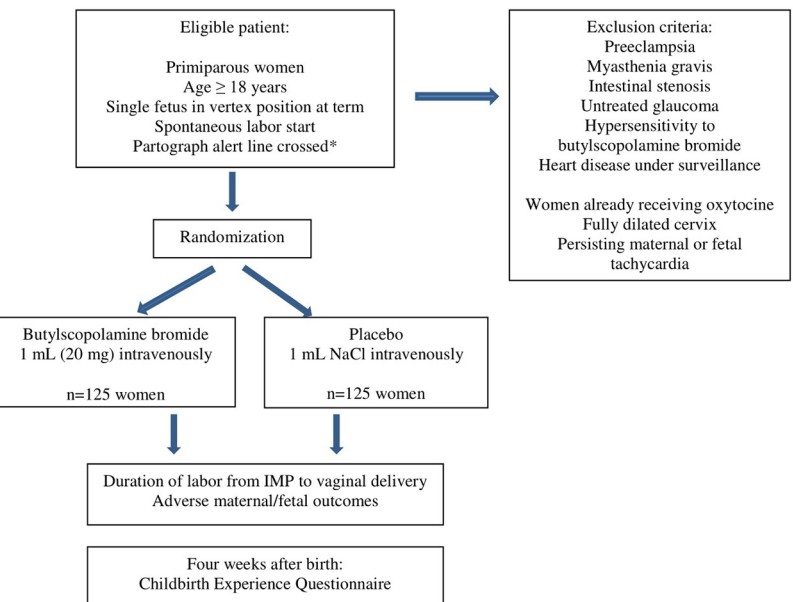

**Fig 2. BUSCLAB trial design.** The flow chart summarizes the design of the BUSCLAB trial. *Crossing of the alert line for slow labor progress, according to the World Health Organization partograph. If the labor curve crosses to the right of the alert line, the cervical dilation rate is <1 cm per hour. NaCl = Sodium Chloride. IMP = Investigational Medicinal Product.

## Interventions

In this study, the IMP is butylscopolamine bromide or placebo (sodium chloride). Women randomized to the intervention group receive 20 mg of butylscopolamine bromide (1 mL Buscopan® from Boehringer Ingelheim (20 mg/mL); 20 mg butylscopolamine bromide, 6 mg sodium chloride, and water) slowly intravenously when they cross the alert line of the WHO partograph. The dose of 20 mg by intravenous route was selected as most previous randomized studies have used this dose and route. When the protocol was written, there was no evidence indicating that a higher dose was more effective. Hence, we chose the lowest effective dose. In addition, the women are given standard care, i.e., augmentation with oxytocin when crossing the partograph action line or due to other reasons, as per ward procedure. If there is indication for butylscopolamine bromide and oxytocin at the same time, both drugs can be given simultaneously. Women randomized to the placebo group receive 1 mL saline solution (Natriumklorid "Fresenius Kabi" 9 mg/mL) slowly intravenously when they cross the alert line of the WHO partograph.

The IMP is given as soon as possible after randomization. If IMP is given more than 45 minutes after the last clinical examination, the midwife examines the cervical dilation again before IMP administration. Given that the medication is administered intravenously as a single dose, lack of compliance is not expected to be a problem. All concomitant medication used by the patient is recorded in the patient's file and case report form (CRF).

We will monitor laboring women and their fetuses after inclusion using fetal cardiotocography (CTG) and measurement of the maternal heart rate for 30 minutes after IMP administration (Fig 1). Adverse effects are registered in the electronic CRF (eCRF). Occurrence, degree, and duration of side effects are registered within 30 minutes after the IMP is given. All women are asked to assess labor pain before and 30 minutes after IMP administration, as well as to fill in a validated questionnaire on birth experience one month after delivery. Data retrieved from

the participants' medical and pregnancy charts including examinations before and during labor are described in Fig 1. We obtain the vital signs from the mother and fetus at baseline, defined as the point of time during labor when slow progress is detected (Fig 1).

## Outcomes

The primary outcome is the duration of labor in minutes from the time when the participant is given IMP to vaginal delivery. Secondary outcomes for the mother include duration of labor from IMP adminstration to 10 cm cervical dilation and from onset of active labor to delivery, the mean cervical dilation rate, mode of delivery, amount of oxytocin given (measured as total time with treatment and in international units [IU]), change in pain score using a visual analogue scale at baseline and 30 minutes after IMP administration, occurrence of postpartum hemorrhage, urinary retention and obstetric anal sphincter injury as well as maternal birth experience. The validated Childbirth Experience Questionnaire that evaluates own capacity, professional support, perceived safety, and participation is distributed to participants four weeks after birth [31]. Secondary outcomes for the infant include Apgar score at 5 and 10 minutes after delivery, pH levels in umbilical vein and artery after delivery, and admission to the neonatal intensive care unit. Further exploratory outcomes include the cervical dilation rate from IMP to first subsequent vaginal exam, indications for operative delivery, CTG patterns, and maternal and fetal heart rates at least 30 minutes after IMP administration.

## Sample-size

In this study, we will include 250 women. The sample-size calculation is based on the randomized controlled trial by Dencker et al. [32] and the review by Neal et al. [7]. Dencker et al. randomized 630 Swedish women with slow progress in the first part of active labor to early versus delayed oxytocin augmentation with similar inclusion criteria and randomization as the BUSCLAB study. In the early oxytocin augmentation group, the mean duration from randomization to delivery was 5.2 (SD 2.8) hours [32]. In the review by Neal et al., 7009 nulliparous women were summarized to have a weighted mean duration of labor of 6 (SD 3.5) hours [7]. The BUSCLAB study defines the active phase of labor from a cervical dilation of 3 cm and measures the duration from IMP administration to delivery. Since the intervention will not start until a woman crosses the alert line, we expect the duration and the corresponding SD to be shorter than result reported in the study by Neal et al. [7], and comparable to results reported by Dencker et al. [32]. We considered a mean difference of 60 minutes in labor duration clinically relevant. Such a difference may be assumed to reduce the use of oxytocin and is realistic to obtain as the mean difference in labor duration between the two treatment groups in the aforementioned studies on antispasmodics was 86 minutes [22].

Our primary outcome (duration from administration of IMP to vaginal delivery) is a time-to-event outcome. Emergency cesarean delivery is a competing event, with an expected occurrence of approximately 8%. With a SD of 2.8 hours and statistical power of 80%, a total of 246 women (123 women in each group) are needed to discover a difference in mean duration of labor of 60 minutes between the two treatment groups. To randomize approximately 250 participants, we estimate an inclusion time of 24–28 months, taking into account non-eligibility of women with planned cesarean delivery or induction of labor, women with spontaneous labor start, but not crossing the alert line, as well as women not willing to participate or not invited to participate due to staff capacity issues.

## Methods: Assignment of interventions

### Randomization

The allocation sequence is computer-generated random numbers. A biostatistician, not involved in inclusion of participants, data handling or data analyses, will assign participants to study drug or placebo by block randomization with randomly mixed block sizes of two, four and six. The allocation ratio is 1:1 (equal probabilities to placebo and treatment). A labor ward midwife enrolls eligible participants. The participants are assigned to treatment based on pre-specified allocation envelopes. The numbered envelopes contain information about what IMP to prepare (butylscopolamine bromide or placebo). A different midwife in the postnatal ward, which is located at a different floor, opens the envelope to reveal the allocated treatment. This midwife prepares the IMP and delivers it to the staff in the labor ward; however, not directly to the midwife taking care of the participant.

### Blinding

Only personnel authorized by the principal investigator (PI) for preparing treatment have access to treatment allocation, and the allocation is not available until the woman in labor has signed the informed consent form and is deemed eligible for the study. All participants and all labor-ward personnel taking care of the participants during labor, the investigators and the statistician are blinded. The midwife in the postnatal ward is the only person not blinded; however, he/she does not provide care to the woman in labor. The placebo and drug preparations are visually identical; both are colorless fluids in 1 mL syringes. The IMPs have standard labeling (including package insert) from the manufacturer until it is prepared by the unblinded study personnel.

## Methods: Data collection, management and analysis

We will perform the data management procedures in accordance with the Norwegian Clinical Research Infrastructures Network (NorCRIN) guidelines (http://www.norcrin.no). The designated investigator staff enters the data required by the protocol into the eCRF using EpiData. Data management personnel perform both manual EpiData review and review of additional electronic edit checks to ensure that the data are complete, consistent, and reasonable. Once the full set of eCRFs have been completed and locked, the sponsor will authorize database lock, and all electronic data will be sent to the designated statistician for analysis. The data will be stored deidentified in a dedicated and secured area at Oslo University Hospital.

### Statistical analysis

The following data sets will be considered for the analyses:

- Intention to treat (ITT) set: Includes all eligible, randomized trial participants with a signed informed consent, regardless of protocol adherence.

- Full analysis (FA) set: Includes all eligible trial participants in the ITT set who have received at least one dose of the IMP.

- Per-protocol (PP) set: Includes all eligible, randomized trial participants with a signed informed consent who have followed the protocol with no major protocol deviations and received one prescribed dose of the same IMP to which they have been randomized.

- Safety set: Includes all eligible, randomized trial participants who have received at least one dose of the IMP, including participants who later revoke their informed consent.

The primary data set for the statistical analysis is the FA set. We will conduct sensitivity analyses based on the ITT and PP sets for comparison reasons.

Prior to the main statistical analysis, we will lock the database for further entering and altering of data. A separate statistical analysis plan will be finalized, signed, and dated prior to database lock and unblinding. The treatment allocation will be revealed after the database lock and used in the statistical analysis. We do not plan to perform interim analyses.

The primary outcome is analyzed using Weibull regression, where we treat emergency delivery as a censoring event, giving a cause-specific hazard ratio of vaginal delivery between the intervention group and the placebo group. Correspondingly, we will also analyze time from IMP administration to cesarean delivery, treating vaginal delivery as a censoring event. The cause-specific hazard ratio of vaginal delivery will be evaluated in light of the cause-specific hazard ratio of cesarean delivery. We will claim superiority of butylscopolamine bromide over placebo on the duration of labor from IMP administration to vaginal delivery if the null hypothesis of no difference between the two treatment groups is rejected at a two-sided significance level of 5% and the treatment difference is in favor of butylscopolamine bromide (i.e., cause-specific hazard ratio of vaginal delivery exceeding 1), given that the cause-specific hazard ratio of cesarean delivery does not indicate increased risk of cesarean delivery in the intervention group. We will check the robustness of the results from the primary analysis by using Cox proportional-hazards regression. We will plot curves of cause-specific cumulative incidence for vaginal delivery and cesarean delivery. In addition, the respective numbers and proportions of women who give birth within 2, 4, 6, 8, 10, and 12 hours after IMP administration will be reported. Time-to-event variables will be analyzed as the primary outcome. We analyze continuous and categorical variables by using linear or median regression and logistic regression. We will summarize patient demographics and baseline characteristics for all participants and by randomization group. We will perform hypothesis generating exploratory analyses on exploratory outcomes (post-hoc analyses). Safety analyses are limited to descriptive statistics and tabulations. Patient-reported outcomes include pain score, side effects and birth experience. We will present all efficacy analyses by the point estimate of the relevant effect measure of the treatment difference, the associated 95% confidence interval, and the *p* value of the corresponding two-sided hypothesis test. We will mainly conduct statistical analyses by using Stata for Windows (StataCorp LLC) and/or R for Windows (R Foundation for Statistical Computing). Some analyses may be conducted by using IBM SPSS Statistics for Windows (IMB Corp.).

## Methods: Monitoring

### Data monitoring

The study is supervised by an external monitor appointed by a clinical study monitor. The monitor performs reviews every six months throughout the study period. Clinical data managers and study monitor may remotely and proactively monitor the eCRFs to improve data quality. The investigator will be visited on a regular basis by the Clinical Study Monitor. The Clinical Trials Unit at Oslo University Hospital is responsible for monitoring the study.

### Adverse events

We will record adverse events (AEs) and serious adverse events (SAEs) for both mother and fetus 30 minutes after IMP administration, on the first day after delivery, and at discharge. The PI is responsible for the detection and documentation of events meeting the criteria and definitions of AEs and SAEs. The main expected AEs in this study are maternal flushing, maternal tachycardia, and fetal tachycardia. An SAE is defined as any untoward medical occurrence that

at any dose results in death of mother or fetus/infant, is immediately life-threatening, requires in-patient hospitalization or prolongation of hospitalization, or results in persistent or significant disability or incapacity. In this study, SAEs include dyspnea, serious skin reactions including Steven-Johnsons syndrome, and anaphylactic reactions including anaphylactic shock. The medical records describe all AEs and SAEs (if any), including causality assessments, the date of and reason for discontinuation from study treatment or withdrawal from the study. Once a year throughout the trial the sponsor will provide the Norwegian Medicines Agency with an annual safety report.

### Emergency unblinding

Unblinding of the treatment allocation is permissible only if the safety and well-being of the participant is compromised. In the event of an SAE, the PI or the attending physician on duty may break the treatment code only if the appropriate management of the patient necessitates immediate knowledge of the current treatment (severe anaphylactic reactions).

### Study management

The PI is responsible for appropriate staff training and task allocation. The investigator is responsible for giving the participating women full and adequate verbal and written information about the nature, purpose, possible risk, and benefit of the study. The PI has insurance coverage for this study through membership of the Norwegian Drug Liability Association.

### Ethics and dissemination

The BUSCLAB trial has EudraCT number 2018-002338-19 and ClinicalTrials.gov identifier NCT03961165. The study protocol is approved by the Regional Committee for Medical and Health Research Ethics North (reference number 2018/2380). The Norwegian Medicines Agency has approved the study (reference number 18/09179-14). Upon study completion and finalization of the study report, we will submit the results of this study for publication and posted in a publicly assessable database of clinical study results. We will also submit the results of this study to the competent authority and the ethics committee according to EU and national regulations.

## Discussion

The BUSCLAB study is powered to clarify whether butylscopolamine bromide has a beneficial role in first-time labor. The protocol has been subject to external reviews by the Norwegian Medicine Agency and the Regional Committee for Medical and Health Research Ethics North in Norway and amended according to their requirements and advice. Given that butylscopolamine bromide has the expected effect, the need for oxytocin to augment contractions may be reduced. Reduced amounts of oxytocin may reduce unwanted side effects. Women who are allocated to the placebo group will contribute to answer an important medical question for future laboring women and their children. Participants need an intravenous access; however, it is routine hospital procedure to establish an intravenous line for the majority of primiparous women due to indications such as epidural analgesia and/or slow labor progress. Women allocated to the intervention group (butylscopolamine bromide) may experience shorter labor if the IMP has the presumed effect. Based on previous studies, the reduction in the active phase of labor will be approximately one hour [22,29]. This reduction will likely take place in the first stage and not in the second stage of labor. There are also possible drug synergies between butylscopolamine bromide and oxytocin on labor progress.

The studies conducted to date seem to have a reassuring safety profile regarding undesirable maternal and neonatal events [22,29]. The drug may be transferred to breast milk; however, full clearance of the drug is believed to take less than 24 hours [25]. Together with the fact that in the first 24 hours after birth the milk production is limited, breastfeeding is not contraindicated in the BUSCLAB trial.

The Cochrane review concluded that no previous RCTs on spasmolytics in labor had reported maternal experience of labor. The BUSCLAB trial aims to use a standardized tool to describe experience of labor in women who receive butylscopolamine bromide compared with placebo.

## Study strengths and limitations

The BUSCLAB study responds to a need for better tools to prevent prolonged labor in first-time mothers at a time when the first signs of slow progress is occurring. The novelty of the research includes assessing prevention of poor labor progress and including an evaluation of the participant's own childbirth experience. The study is, however, restricted to primiparous women and does not assess the effect of butylscopolamine bromide in parous women. The study does not assess the effect of different doses of IMP or the effect of repeated IMP doses in long labors and is not powered to detect differences in mode of delivery or maternal or perinatal morbidity.

## Status and timeline of the study

The study started inclusion on June 1, 2019, with a planned inclusion over 24–28 months. End of the trial is defined as when the Childbirth Experience Questionnaire has been received from all participants and no later than two months after the last patient is included.

## Supporting information

**S1 File. Spirit vhecklist for the BUSCLAB study.**
(DOC)

**S2 File. BUSCLAB patient consent form.**
(PDF)

**S3 File. Buscopan protocol version 14.**
(PDF)

**S4 File. Funding_documentation from Stiftelsen Dam/NKSF.**
(PDF)

**S5 File. Funding_documentation from the Royal Norwegian Society of Sciences and Letters.**
(PDF)

## Acknowledgments

We wish to thank the staff at the labor and postnatal ward at Oslo University Hospital Rikshospitalet.

## Author Contributions

**Conceptualization:** Lise Christine Gaudernack, Trond Melbye Michelsen.

**Funding acquisition:** Trond Melbye Michelsen.

**Investigation:** Ingvil Krarup Sørbye.

**Methodology:** Ingvil Krarup Sørbye, Lise Christine Gaudernack, Leiv Arne Rosseland, Mirjam Lukasse, Nina Gunnes, Trond Melbye Michelsen.

**Project administration:** Trond Melbye Michelsen.

**Resources:** Trond Melbye Michelsen.

**Supervision:** Ingvil Krarup Sørbye, Mirjam Lukasse, Trond Melbye Michelsen.

**Writing – original draft:** Ingvil Krarup Sørbye, Lise Christine Gaudernack, Nina Gunnes, Trond Melbye Michelsen.

**Writing – review & editing:** Ingvil Krarup Sørbye, Lise Christine Gaudernack, Angeline Einarsen, Leiv Arne Rosseland, Mirjam Lukasse, Nina Gunnes, Trond Melbye Michelsen.

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
