## [Decision Letter · Decision Letter 0]

11 Jul 2022

PONE-D-21-38415Study protocol for the BUSCopan in LABor (BUSCLAB) study: A randomized placebo-controlled trial investigating the effect of butylscopolamine bromide to prevent prolonged laborPLOS ONE

Dear Dr. Sørbye,

Thank you for submitting your manuscript to PLOS ONE. After careful consideration, we feel that it has merit but does not fully meet PLOS ONE’s publication criteria as it currently stands. Therefore, we invite you to submit a revised version of the manuscript that addresses the points raised during the review process.

Specifically, the reviewers have concerns about the rationale of the proposed study and request more details on your methodology. Please have all the comments addressed point-by-point.

We look forward to receiving your revised manuscript.

Kind regards,

Jianhong Zhou

Staff Editor

PLOS ONE

Journal Requirements:

https://journals.plos.org/plosone/s/file?id=ba62/PLOSOne_formatting_sample_title_authors_affiliations.pdf,

2. Please include a copy of Table 1 which you refer to in your text on page 13.

Reviewers' comments:

Reviewer's Responses to Questions

**Comments to the Author**

1. Does the manuscript provide a valid rationale for the proposed study, with clearly identified and justified research questions?

Reviewer #1: Partly

Reviewer #2: Yes

Reviewer #3: Yes

2. Is the protocol technically sound and planned in a manner that will lead to a meaningful outcome and allow testing the stated hypotheses?

Reviewer #1: Yes

Reviewer #2: Yes

Reviewer #3: Yes

3. Is the methodology feasible and described in sufficient detail to allow the work to be replicable?

Reviewer #1: Yes

Reviewer #2: Yes

Reviewer #3: Yes

4. Have the authors described where all data underlying the findings will be made available when the study is complete?

Reviewer #1: No

Reviewer #2: No

Reviewer #3: Yes

5. Is the manuscript presented in an intelligible fashion and written in standard English?

Reviewer #1: No

Reviewer #2: No

Reviewer #3: Yes

6. Review Comments to the Author

You may also provide optional suggestions and comments to authors that they might find helpful in planning their study.

Reviewer #1: This study focus on prolonged labour and the use of buscopan.

The study aims to evaluate the duration of labour when using buscopan for prolonged labour comped to placebo.

General:

Why is it important to reduce duration of labour? Why is this the most important outcome, the primary outcome of your study?

Does the women prefer a 60 minutes reduction? Does the clinicians?

Why is the study updated as completed at clinicaltrials.gov? Have you already terminated the inclusion?

Consider using SPIRIT checklist when presenting a trial protocol – the chronology of paper could be improved by the use of spirit.

You repeat yourself several times during the paper. Please do only report eg. the objective and the outcome measures once.

Try to be more concise – current the word count is very high.

Introduction:

A definition of prolonged labour is lacking, also definition of active phase should be included in the introduction.

“Prolonged labour is common”, could you quantify?

Oxytocin- suggest that you focus on the primary adverse effect of oxytocin: uterine hyperstimaultion (tacycysystole) which may lead to compromised oxygen supply for the baby, fetal asphyxia and the need for immediate delivery. Other adverse effects could also be mentioned (postpartum hemorrhage, water intoxication, uterine rupture). The effect on breastfeeding remains as far as I am aware still unknown.

Instead of criticizing oxytocin you might consider focusing on the synergistic effect of two different drugs?

Prolonged labour – should also lead to the considerations as to whether there are other explanations as to whether labour do not progress – e.g. fetal/pelvic disproportion

Study overview:

Your hypothesis and objective should be merged into one sentence. And comnsider only to report the primary objective of the trial (like a PICO presented as a sentence)

You should report primary outcome and secondary outcomes, not ‘objectives’

Patient enrollment and eligibility criteria:

Please define the active phase for your study earlier in the text or at least mention in the introduction the variety of different definition. Do you only consider cervical dilatation to be relevant, not contractions, amniotomy, cervical length?

Please consider moving ref 26 and 27 to the introduction

No need to both text the inclusion criteria and exclusion criteria and also put the in a box. Choose one of the two options.

Sample size calculation:

Consider a brief presentation of the Dencker trial in order to help the reader, they randomised, but to which intervention?

Please consider to describe your samplesize calculation with less word, more consiese.

Is the trial GCP monitored? If yes please report that, if not please argue why not?

Reviewer #2: The manuscript needs a language editor. There are so many typographical errors.

ABSTRACT

The objectives of the study is not well defined.

The authors stated: Women are

included in the first stage of labor if they cross the partograph alert line with a cervical

dilation between 3–9 cm. Does it mean that authors recruit patients at 9cm cervical dilatation? If yes, would that not be too late?

INTRODUCTION

The authors stated: Another class of drugs, antispasmodics, has been used to prevent poor progress of labor in both

developing and developed countries. The authors should replace developed to 'high-income' and replace developing to 'low and middle-income'

Reviewer #3: Comments to the Author

I read with great interest the enclosed Manuscript which falls within the aim of Plos One.

In my honest opinion, the topic is interesting enough to attract the readers’ attention. Methodology is accurate.

However, it is necessary to cite and discuss previous trials and meta-analyses regarding the aim of trial, since there are several trials on the issue. See and cite this updated meta-analysis which summarizes the available evidence from RCTs, PMID: 32629224.

7. PLOS authors have the option to publish the peer review history of their article (what does this mean?). If published, this will include your full peer review and any attached files.

Reviewer #1: No

Reviewer #2: No

Reviewer #3: No

---

## [Author Response · Author response to Decision Letter 0]

3 Aug 2022

To: Staff Editor Jianhong Zhou 

PLOS ONE 

Resubmission date: August 3rd, 2022

Original Submission date: December 4th 2021

Dear Dr. Jianhong Zhou, 

Thank you for the thorough review from Editor and reviewers considering our submitted manuscript entitled: “Study protocol for the BUSCopan in LABor (BUSCLAB) study: A randomized placebo-controlled trial investigating the effect of butylscopolamine bromide to prevent prolonged labor”.

We have revised the manuscript according to the valuable suggestions made and we believe the manuscript has improved considerably.

Our responses to the comments are enclosed in the attachment "Response to Reviewers".

The word count is now 4072 words.

Best regards,

Dr. Ingvil Krarup Sørbye, corresponding author

---

## [Decision Letter · Decision Letter 1]

11 Oct 2022

Study protocol for the BUSCopan in LABor (BUSCLAB) study: A randomized placebo-controlled trial investigating the effect of butylscopolamine bromide to prevent prolonged labor

PONE-D-21-38415R1

Dear Dr. Sørbye,

We’re pleased to inform you that your manuscript has been judged scientifically suitable for publication and will be formally accepted for publication once it meets all outstanding technical requirements.

Kind regards,

Hanna Landenmark

Staff Editor

PLOS ONE

Additional Editor Comments (optional):

Reviewers' comments:

Reviewer's Responses to Questions

**Comments to the Author**

1. Does the manuscript provide a valid rationale for the proposed study, with clearly identified and justified research questions?

Reviewer #1: Yes

Reviewer #2: Yes

Reviewer #3: Yes

2. Is the protocol technically sound and planned in a manner that will lead to a meaningful outcome and allow testing the stated hypotheses?

Reviewer #1: Yes

Reviewer #2: Partly

Reviewer #3: Yes

3. Is the methodology feasible and described in sufficient detail to allow the work to be replicable?

Reviewer #1: Yes

Reviewer #2: No

Reviewer #3: Yes

4. Have the authors described where all data underlying the findings will be made available when the study is complete?

Reviewer #1: No

Reviewer #2: No

Reviewer #3: Yes

5. Is the manuscript presented in an intelligible fashion and written in standard English?

Reviewer #1: Yes

Reviewer #2: No

Reviewer #3: Yes

6. Review Comments to the Author

You may also provide optional suggestions and comments to authors that they might find helpful in planning their study.

Reviewer #1: Congratulation, a interesting RCT is coming up. The paper has been improved and I find it ready for publication,

Reviewer #2: The authors did not address all the issues previously raised. If they addressed it, the authors must outline/highlight it in yellow. They need to be detailed on their specific revision

Reviewer #3: Thank you for giving me the opportunity of reviewing the revised version of the enclosed manuscript. All the issues highlighted in the previous version were solved by the authors. No further concerns.

7. PLOS authors have the option to publish the peer review history of their article (what does this mean?). If published, this will include your full peer review and any attached files.

Reviewer #1: No

Reviewer #2: **Yes: **George

Reviewer #3: No

---

## [Editor Report · Acceptance letter]

25 Oct 2022

PONE-D-21-38415R1 

Study protocol for the BUSCopan in LABor (BUSCLAB) study: A randomized placebo-controlled trial investigating the effect of butylscopolamine bromide to prevent prolonged labor 

Dear Dr. Sørbye:

I'm pleased to inform you that your manuscript has been deemed suitable for publication in PLOS ONE. Congratulations! Your manuscript is now with our production department. 

Kind regards, 

on behalf of

Dr. Hanna Landenmark 

Staff Editor

PLOS ONE